# An analysis of the impact of China's macroeconomic performance on its trade partners: Evidence based on the GVAR model

Aftab Alam[1], Jingmei Ma[1]*, Ibrar Hussain[2]*, Rizwan Fazal[3]

**1** Department of Economics and Management, Harbin University of Science and Technology, Heilongjiang, China, **2** Department of Economics, University of Malakand, Chakdara, Dir Lower, Khyber Pakhtunkhwa, Pakistan, **3** United Nations Development Program (UNDP), Sareena Business Complex, Islamabad, Pakistan

☯ These authors contributed equally to this work.
\* maggiesisu@163.com (JM); ibrarhussain_uom@yahoo.com (IH)

## Abstract

Economic strategies and planning are critical to a country's growth and development. China, like many other countries, is seeking the most cost-effective trade deals. Using the Global Vector Auto Regression (GVAR) model, this study examined the impact of a shock to China's macroeconomic factors on trading economies. The major findings reveal that there is no co-movement between the shock in Chinese gross domestic product (GDP) and German macroeconomic indicators; however, the shock has a positive and substantial influence on Japan's GDP and Unites States (US)' exchange rate. It is also worth noting that a shock to Chinese trade volume is more susceptible and more disturbing than a shock to US trade volume since it reduces trade volume and causes the Ren Min Bi (RMB) to devalue permanently. Furthermore, the analysis shows that Chinese stock prices have a major influence on German economy since China's GDP, trade volume, and currency appreciate over time when its stock price rises. Finally, the exchange rate shock is beneficial to Germany as it boosts GDP and trade volume but has a negative influence on US stock prices. The current study is, therefore, expected to be a suitable beginning point for the governments and policy-makers of trading partners to design an effective trade policy to minimize the impact on major economic variables.

## 1. Introduction

Like many other countries, China is looking for the most cost-effective way to trade with countries worldwide. Historically, China used to trade oil, using the Strait of Malacca as the only route available to them. But, the face of global politics is changing, and due to international politics and conflicts, China is looking for other alternatives to be more safe and convenient for trade [1]. China has been restructuring the international trade pattern since the 1970s and is now the world's second largest economy by nominal GDP. Global trade increased from $6 trillion in 1990 to $37 trillion in 2014, and China is among the largest exporters of world trade, so the contribution to the world GDP is inevitable. One may argue that trade with China's

**Funding:** This work was supported by the National Natural Science Foundation of China [No. 71972063, 71672051].

**Competing interests:** he authors have declared that no competing interests exist.

rapidly growing economy may not always be harmful to its trading partners' economies. However, China's growth poses a challenge to the Organization for Economic Cooperation and Development (OECD)'s Western economies, notably the United States [2]. The fact that China maintains a de facto fixed exchange rate regime, with the RMB tied to the US dollar within a restricted trading band, has gotten a lot of attention recently [3]. Among others, they claim that China has intentionally weakened its currency to benefit from increased exports. Lower Yuan exchange rates improve external demand for Chinese goods while lowering demand from China's trade partners, such as the United States, Germany, and Japan. The concern now is whether China's currency rate fluctuations have a large or minor influence on its trading partners [3, 4]. This study employs the Global Vector Auto Regression (GVAR) model of Pesaran, Schuermann, and Weiner [5] to examine and quantify the impact of the Chinese economic shock on the macroeconomic variables of its major trading partners (US, Japan, and Germany) in terms of an increase in real GDP, real equity prices, trade volume, and exchange rate. The significance and novelty of this study stem from the fact that it is the first to use the newly developed GVAR model to investigate the impact of shocks in the Chinese economy (exchange rate, trade volume, equity prices, and GDP) on its trading partners. The findings of the study should also have a significant influence on economic integration strategies and management. The current analysis will enable the government and trading officials to develop efficient trade policies to mitigate and quantify the impact of Chinese economic shocks. The remainder of this paper consists of four sections. Section 2 discusses materials and methods; Section 3 presents results and discussion; and Section 4 sheds light on the conclusion and policy implications.

## 1.1 China-US trade agreement and its divergent effect on trading partners

The China-US Economic and Trade Agreement was recently signed in 2020, with China agreeing to halve tariffs on 1,717 US items such as soybeans and crude oil that were imposed in 2018 [6]. The US is also expected to lower duties on certain Chinese imports in lockstep. Tables 1 and 2 illustrate China's imports and exports of products from its major trade partners in 2018.

China clearly imports a lot of products from these countries, such as crude oil worth $20.67 billion from Russia, industrial machinery worth $20.83 billion from Germany, and oilseeds totaling $20.32 billion from Brazil [7]. Since these products are also covered by the Economic and Trade Agreement, there must be trade diversion impacts on foreign trading partners if China redirects its imports to the US to meet purchasing commitments. Due to reduced soybean shipments, Brazil will be the most affected among China's key trading partners in the agriculture sector. Brazil is the world's largest exporter of soybeans, with China accounting for roughly 80% of those shipments in 2017. Other developing nations, in addition to Brazil, are impacted by the trade agreement between the US and China [8].

## 1.2 China's major trading partners

Table 3 depicts China's trade relations with its trading partners. According to Chinese trade data from 2018 (total trade volume), the mentioned trading partners include 28 European Union (EU) countries, ten Association of Southeast Asian Nations (ASEAN) nations (Japan, South Korea, Hong Kong, and Taiwan), and the United States.

According to trade statistics, the United States, the EU, and ASEAN were China's top three export markets, while the EU, ASEAN, and South Korea were its main import sources. China has large trade surpluses with the US ($282 billion), Hong Kong ($274 billion), and the European Union ($129 billion), but large trade deficits with Taiwan ($112 billion) and South Korea ($74 billion). However, China's trade statistics differ greatly from those of the United States

**Table 1. China imports (2018) in US billion $.**

| Category | Russia | Australia | Brazil | Japan | US |
|---|---|---|---|---|---|
| **AGRICULTURTE** | 1.78 | 8.65 | 23.22 | 0.56 | 212 |
| Cereals | 0.04 | 1.38 | 0.00 | 0.00 | 1.56 |
| Cotton | 0.00 | 0.35 | 0.13 | 0.00 | 0.98 |
| Meat | 0.00 | 1.02 | 1.79 | 0.00 | 0.57 |
| Oil seeds | 0.14 | 0.00 | 20.32 | 0.00 | 12.4 |
| Other agriculture commodities | 0.52 | 5.62 | 0.96 | 0.44 | 4.55 |
| Seafood | 1.09 | 0.28 | 0.01 | 0.11 | 1.25 |
| **MANUFACTURING** | 6.20 | 3.12 | 1.98 | 82.9 | 66.8 |
| Aircraft | 0.12 | 0.00 | 0.42 | 0.00 | 13.3 |
| Electrical equipment and machinery | 0.59 | 0.33 | 0.08 | 21.3 | 6.26 |
| Industrial machinery | 1.50 | 0.23 | 0.43 | 31.0 | 15.0 |
| Iron and steel | 0.05 | 0.02 | 0.60 | 5.86 | 0.71 |
| Optical and medical instruments | 0.01 | 0.17 | 0.01 | 2.00 | 4.06 |
| Other manufactured goods | 3.81 | 1.73 | 0.43 | 15.5 | 12.9 |
| Pharmaceutical products | 0.00 | 0.63 | 0.02 | 0.77 | 3.38 |
| Vehicles | 0.12 | 0.00 | 0.00 | 6.40 | 11.0 |
| **ENERGY** | 22.5 | 15.2 | 7.37 | 0.02 | 6.99 |
| Coal | 1.77 | 8.87 | 0.00 | 0.00 | 0.40 |
| Crude oil | 20.67 | 0.53 | 7.37 | 0.00 | 4.09 |
| LNG | 0.04 | 5.72 | 0.00 | 0.00 | 0.47 |
| Refined products | 0.02 | 0.11 | 0.00 | 0.02 | 2.03 |
| **Total** | **30.48** | **26.9** | **32.57** | **83.5** | **95.1** |

Source: China's Customs Administration

[9]. Further, Fig 1 depicts China's total volume of trade and its split into exports and imports with the major trade partners.

## 1.3 China's exchange rate and foreign trade policy

For the past few years, China has primarily fixed the RMB against the US dollar [10], with the central bank publishing a central parity rate (CPR) of exchange between the RMB and the US

**Table 2. China exports (2018) in US billion $.**

| Category | US | Japan | Germany | Australia |
|---|---|---|---|---|
| **Electrical, electronic equipment** | 199.51 | 35.28 | 18.4 | 8.61 |
| **Machinery, nuclear reactors, boilers** | 102.99 | 25.19 | 17.2 | 8.35 |
| **Furniture, lighting signs, prefabricated buildings** | 33.38 | 8.02 | 4.05 | 3.25 |
| **Toys, games, sports requisites** | 19.45 | 7.50 | 3.56 | 2.29 |
| **Plastics** | 18.46 | 4.80 | 3.24 | 2.17 |
| **Vehicles other than railway, tramway** | 18.08 | 4.63 | 2.71 | 1.98 |
| **Articles of apparel knit or crocheted** | 17.65 | 4.45 | 2.62 | 1.86 |
| **Articles of apparel ,not knit or crocheted** | 14.56 | 4.44 | 2.63 | 1.77 |
| **Footwear, gaiters and the like** | 12.14 | 3.63 | 2.00 | 1.39 |
| **Articles of iron or steel** | 11.86 | 3.53 | 2.00 | 1.21 |

Source: China's Customs Administration

**Table 3. China's major merchandize trading partners in 2018 ($ billions).**

| Country | Total Trade | Chinese Exports | Chinese Imports | Trade Balance |
|---|---|---|---|---|
| European Union | 681 | 408 | 273 | 135 |
| United States | 631 | 477 | 154 | 323 |
| ASEAN | 575 | 318 | 257 | 61 |
| Japan | 327 | 147 | 180 | -33 |
| South Korea | 313 | 109 | 204 | -95 |
| Hong Kong | 310 | 302 | 8 | 294 |
| Taiwan | 225 | 48 | 177 | -129 |

Source: China's Customs Administration. Notes: Rankings according to China's total trade in 2018. China's bilateral trade data often differ from that of its trading partners.

dollar on a daily basis. The Chinese government's central exchange rate with the US dollar was 8.28 RMB per dollar on average in 1998, and this rate stayed stable until 2005 [3, 11]. However, under pressure from trading partners such as the United States, China promised to increase the RMB's value by 2.1 percent over the next three years by pegging its currency to a basket of currencies [4]. Due to the global economic crisis, China suspended RMB appreciation in 2008 in order to reduce the impact of financial problems by expanding exports. The RMB appreciated 35.3 percent against the US dollar between 2005 and 2015. The Chinese central bank declared in 2015 that the daily RMB central parity rate would become more "market-oriented."

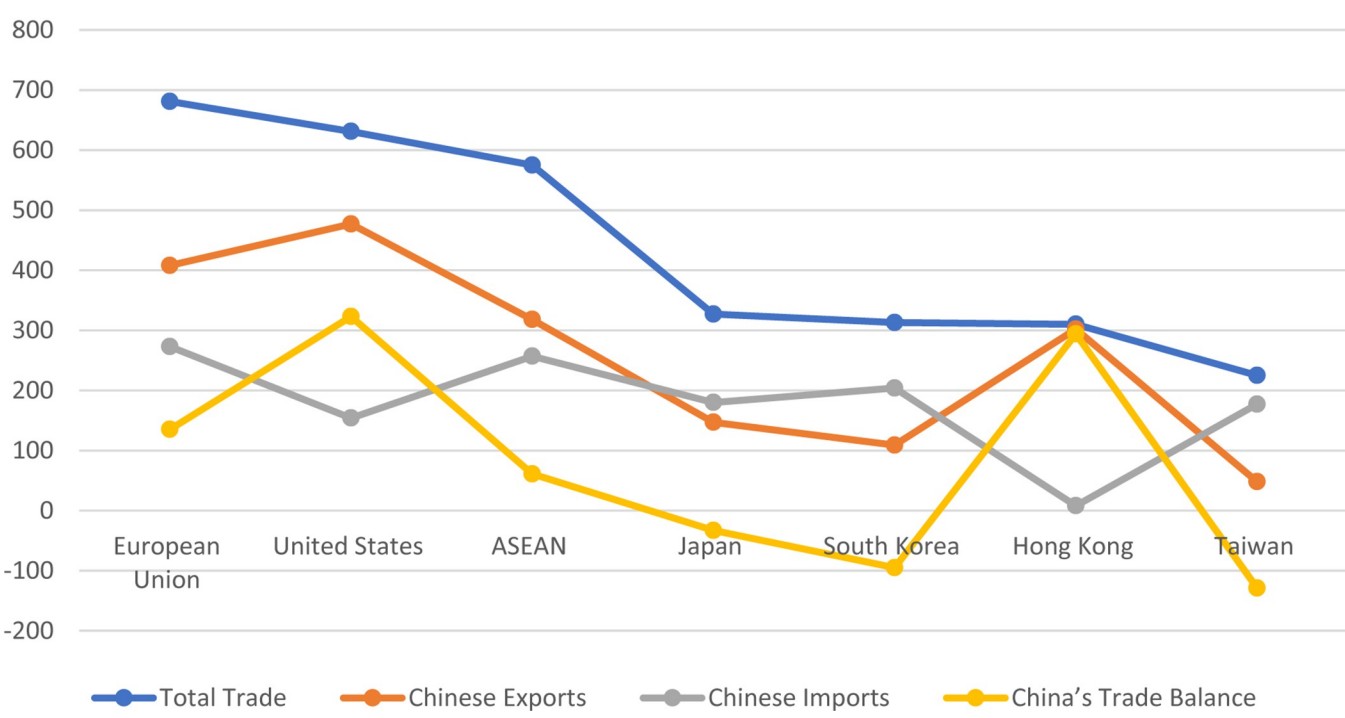

**Fig 1. China's major merchandize trading partners in 2018.** According to trade statistics, the United States, the EU, and ASEAN were China's top three export markets, while the EU, ASEAN, and South Korea were its main import sources. China has large trade surpluses with the US ($282 billion), Hong Kong ($274 billion), and the European Union ($129 billion), but large trade deficits with Taiwan ($112 billion) and South Korea ($74 billion). However, China's trade statistics differ greatly from those of the United States.

The RMB, on the other hand, fell by 4.4 percent against the dollar in the three days that followed, and it continued to fall against the dollar for the rest of 2015 and into 2016. Between August 2015 and December 2016, the RMB lost 8.8% of its value against the U.S. dollar. Between January and December 2017, the RMB appreciated by 4.6 percent against the USD. The RMB's value versus the dollar has been trending lower since 2018.

Many experts believe the RMB's recent depreciation is partly due to China's weakening economy and the uncertainty created by the ongoing US-China trade war. Due to the RMB's 8.2 percent depreciation from March 2018 to May 2019, the US administration boosted tariffs on Chinese imports from 10% to 25% in May 2019, which will have a detrimental impact on China's economy. Others claim that China has interfered in currency markets to depreciate the RMB in order to mitigate the economic impact of US tariff rises. China surpassed Germany in 2009 to become the world's largest merchandise exporter as well as the world's second-largest merchandize importer after the United States. However, China became the world's largest merchandise trading (exports plus imports) economy in 2012, surpassing the United States. China's proportion of worldwide merchandize exports increased from 2.0% in 1990 to 14.1 percent in 2015, but then dropped to 13.4% in 2016 and 13.2% in 2017.

In a study, Xu [12] examines in detail the fundamentals that determine the exchange rate in China and comes up with two important findings in this regard. The first finding indicates that during the last few decades, China has determined adequate local market prices that are connected to global market prices. As a result, the exchange rate can act as a useful nominal anchor. In terms of exchange rate stability, domestic prices remain steady. Another finding shows that changes in the exchange rate have only a limited impact on trade flows. For current account balances, exchange rate flexibility is not required. As a result, the data suggests that China will stick to its strategy of maintaining exchange rate stability. Galesi and Lombardi [13] investigated the influence of China's exchange rate policies on trade with Asian countries. The long-run relationship between variables was found by using the cointegration technique. In addition, the reduced form export and import equations were applied to avoid the simultaneous equation bias that would occur if the supply and demand functions were estimated independently. The study indicates that appreciation of the RMB had a negative influence on overall exports to Asian countries. Ahmed [14] investigates the sensitivity of Chinese exports due to changes in exchange rates for the period 1997 to 2004. The author developed two types of export models, one for processed export and the other for non-processed export. The findings show that devaluation reduces exports of both processed and non-processed goods. Non-processing exports are notably negative, whereas processing exports are positive but insignificant. As a result, an increase in China's exchange rate reduces the country's trade surplus. For quarterly data from 1995Q1 to 2011Q4, Cardoso and Duarte [15] evaluated the influence of Chinese exchange rate policy on world trade with the European Union. The findings indicate that Chinese exports increased as a result of a currency intervention, which occurs when a government or central bank buys or sells foreign currency in exchange for its own domestic currency.

The literature review revealed that China's exchange rate is complex, necessitating deeper investigation into trade linkages with currency rates as well as an examination of risk and uncertainty for its trading partners. Several studies on China's currency depreciation and trade policies, as well as their impact on trading partners, are presented in Table 4, with mixed results, indicating a lack of consensus.

It is critical to comprehend China's interactions with its trading partners in order to build effective trade policies to limit the impact of Chinese economic shocks. Therefore, the current study is designed to respond to shocks in China's macroeconomic variables and their influence on its trade partners.

**Table 4. Literature on China exchange rate and trade.**

| Study | Methodology | Findings |
|---|---|---|
| Cerra and Dayal-Gulati [16] | Error Correction Model | For the years 1983 to 1997, there were no effects on exports or imports. However, for the years 1988 to 1997, there was a negative and significant impact on exports as well as a positive and significant mixed impact. |
| Cerra and Saxena [17] | Dynamic OLS | Export price elasticity increases at the end of the period, while the nominal effective exchange rate (NEER) has no significant impact at all. |
| Eckaus [18] | OLS | There is a significant negative effect on exports to the US and China's share of US imports. Therefore, China should appreciate its currency. |
| Marquez and Schindler [19] | OLS | An appreciation lowers ordinary imports; the effect is robust for processed imports, but no such impact was observed in the case of exports. |
| Li, Voon, and Ran [20] | OLS | The negative impact of currency appreciation on exports was observed. |
| Bénassy-Quéré and Lahrèche-Révil [21] | Gravity Model | RMB real depreciation increases China's exports to the OECD while reducing Asian exports to China. |
| Cardoso and Duarte [15] | VECM | Over the past few years, Chinese exports have benefited from an unfair competitive advantage resulting from the manipulation of its currency value. |
| Steinberg and Tan [8] | Gravity Model | East Asian currencies would benefit Asia and the rest of the world if they could appreciate together against external currencies while maintaining relative currency stability within the region. |

## 2. Materials and methods

There are various tools available for capturing shocks in macroeconomic variables in one economy and their impact on other economies. However, this study applied the Global Vector Autoregressive (GVAR) methodology developed by Pesaran et al. [5], which offers a general as well as a practical global modelling framework and has proven to be a very useful approach to analyze interactions in the global macro economy and other data networks where both the cross-section and the time dimensions are large. The application of GVAR is suitable for the justification of many transmission mechanisms like monetary policy, exchange rate, economic slowdown, the role of financial markets in the transmission of international business cycles and international inflation inter-linkages. The GVAR methodology consists of two steps. In the first step, it uses the domestic macroeconomic, country-specific foreign, and global variables and estimates a Vector Autoregressive (VAR) for each country. In the second step, a GVAR is constructed from the estimated country-specific models, which is used to generate the generalized impulse response functions (GIRFs). The GVAR methodology assumes that in country-specific models, external variables perform weakly exogenously, and the structural stability of the country-specific models is based on test statistics.

### 2.1 Structure of the standard GVAR model

As discussed earlier, the GVAR model is not estimated at once but estimated on the basis of each individual country-specific VAR model. The standard VAR model is a general form of the univariate autoregressive model (AR model) by allowing more than one variable. It is the stochastic process used to capture the linear interdependencies among a set of a country's $i's$ macroeconomic variables $x_{it}$. The VAR (p) for the country $i$ is presented as:

$$x_{it} = \alpha_{i0} + \beta_i \sum_{j=1}^{p} x_{i,t-j} + \varepsilon_{it} \qquad (1)$$

$i = $ ith countries$t = 1, 2 \ldots \text{T}$

In Eq (1), $\alpha$ and $\beta$ are the coefficient vectors and $\varepsilon_{it}$ is assumed independent identically distributed i.e. $\varepsilon_{it} \sim i.i.d.\ (0, \Sigma ii)$. $x_{it}$ is a vector of endogenous domestic variables in our case it is consisting of the real exchange rate ($EX_{it}$), trade volume ($T_{it}$), real equity price ($Q_{it}$), and real GDP ($Y_{it}$). In vector form $x_{it} = (T_{it}, Q_{it}, EX_{it}, Y_{it})'$.

## 2.2 Country specific VARX models

VARX is the extension of the VAR (p) model that includes the country-specific foreign variables $x_{it}^*$. The external variables are calculated by the weighted sum of specific variables. The weight $w_{ij}$ is the bilateral trade weight of country $i$ in country $j$, therefore $w_{it} = 0$ and $\sum_{i=1}^{n} w_{ij} = 1$. There weights will also be used for the computation of the foreign variables $w_{ij}$ is separating the share of country $j$ in the trade (export plus imports) of country $i$. The country level trade share is constructed by dividing the total trade of each country $i$ by the amount of trade with country $j$, such the $i_{th}$ row sums to one, for all the $i$. $x_{it}^*$, the vector of weak exogenous external variables is expressed as:

$$x_{it}^* = \left(T_{it}^*, Q_{it}^*, EX_{it}^*, Y_{it}^*\right)'.$$

Where:

$$T_{it}^* = \sum_{j=0}^{N} w_{ij} T_{jt}, \ \ Q_{it}^* = \sum_{j=0}^{N} w_{ij} Q_{jt}, \ \ EX_{it}^* = \sum_{j=0}^{N} w_{ij} EX_{jt}, \ \ Y_{it}^* = \sum_{j=0}^{N} w_{ij} Y_{jt}$$

The VARX $(p, q)$ for country $i$ is presented as,

$$x_{it} = \alpha_{i0} + \beta_i \sum_{j=1}^{p} x_{i,t-j} + \gamma_{i0} \sum_{j=0}^{q} x_{it-j}^* + \varepsilon_{it} \tag{2}$$

Where;
$\beta_i = k{\times}k$ Matrix of coefficient associated with lagged domestic variables.
$\gamma_{i0} = k{\times}k^*$ *Coefficient matrix of foreign specific variables*
$\varepsilon_i = k{\times}1$ Vectors denotes concerned country oriented disturbances shocks
In order to obtain the VECM for country $i$, for simplicity consider VARX (1, 1) as:

$$x_{it} = \alpha_{i0} + \beta x_{i,t-1} + \gamma_{i0} x_{it}^* + \gamma_{i1} x_{i,t-1}^* + \varepsilon_{it}$$

The above VAR (1, 1) model is then written in error correction form as:

$$\Delta x_{it} = \alpha ECM_{i,t-1} + \gamma_{i0} \Delta x_{it}^* + \varepsilon_{it} \tag{3}$$

Where;

$$ECM_{i,t-1} = \alpha\left(\beta_{ix}^{\circ} x_{i,t-1} + \beta_{ix}^{\circ} x_{i,t-1}^*\right)$$

$ECM_{i,t-1}$ is a vector of long run co-integrating relations, also known as error correction terms corresponding to the $r_i$ cointegrating relations of the $i^{th}$ country model. The rank of $\alpha\beta_i^{\circ}$ is determined using the maximum Eigen value or the trace statistics. $\beta_i^{\circ}$ is estimated by imposing suitable exact or possibly over identifying restrictions on the elements of $\beta_i^{\circ}$.

## 2.3 Testing for weak exogeneity of foreign variables

The GVAR methodology assumes that in country-specific VARX $(p, q)$ models the external variables $x_{it}^*$ perform weakly exogenous. For an individual country, foreign variables and global variables are constructed along the lines described by Johansen [22] and Harbo, Johansen,

Nielsen, and Rahbek [23] for the test of focused countries. To verify the weak exogeneity of $x_{it}^*$ with respect to the domestic variables, following auxiliary regression for each $l^{th}$, the element of foreign variables $x_{it}^*$ will be estimated:

$$\Delta x_{it}^* = \mu_{il} + \sum_{j=1}^{r_i} \theta_{ij,l} ECM_{i,t-1}^j + \sum_{k=1}^{s_i} \sigma_{ik,l} \Delta x_{i,t-k} + \sum_{m=1}^{n_i} \vartheta_{im,l} x_{i,t-m}^* + \varepsilon_{it,l} \tag{4}$$

Test the joint significance of the coefficients on the estimated error correction terms as:

$$H_O : \theta_{ij,l} = 0; \qquad\qquad \text{For } j = 1, 2, 3 \ldots \ldots . \theta_i$$

If the F-test cannot reject the null hypothesis so the assumption of weak exogeneity is satisfied. Weak Exogeneity test guides the specification of country-specific models.

## 2.4 The GVAR model

The second step of GVAR is to arrange the estimated country models in order to get the global VAR model. Therefore, rewriting the country-specific VARX equation as:

$$A_i Y_{it} = \alpha_{i0} + \pi_i Y_{i,t-i} + \varepsilon_i \tag{5}$$

Where $Y_{it}$ is a vector of both $x_{it}$ and $x_{it}^*$; $Y_{it} = (x_{it}, x_{it}^*)$, $A_i = (I_{ki}, \gamma_{i0})$ Having dimension $I \times (k \times k^*)$ and $\pi_i = (\beta_i, 0)$ matrix having dimension: $\pi_i(k \times k^*)$.

As we know that $\sum_{j=0}^{N} w_{ij} x_i = x_{it}^*$ the $Y_{it}$ can be written as:

$$Y_{it} = w_i x_i.$$

$w_i$ Is a $k(k \times k^*)$ link matrix which is consists of ones and country specific weight i.e.:

$$w = \begin{pmatrix} w_1 & 0 & 0 & 0 \\ 0 & w_2 & 0 & 0 \\ 0 & 0 & . & 0 \\ & & & . \\ & & & . \\ 0 & 0 & & w_N \end{pmatrix}$$

In terms of link matrix Eq (5) can be written as:

$$A w_i x_i = \alpha_{i0} + \pi_i w_i x_{t-1} + \varepsilon_{it} \tag{6}$$

Rewrite Eq 6 for every country the global VAR model are as follow:

$$H x_t = \alpha_0 + M x_{t-1} + \varepsilon_t \tag{7}$$

Whereas

$$H = \begin{pmatrix} A_0 W_0 \\ A_1 W_1 \\ . \\ . \\ A_N W_N \end{pmatrix}, \alpha_0 = \begin{pmatrix} \alpha_{00} \\ \alpha_{10} \\ . \\ . \\ \alpha_{N0} \end{pmatrix} M = \begin{pmatrix} \pi_0 W_0 \\ \pi_1 W_1 \\ . \\ . \\ \pi_N W_N \end{pmatrix}$$

Matrix H is full ranked so the global solution can be written as:

$$x_t = b_{0,t} + K x_{t-1} + \mu_t \tag{8}$$

Eq 8 is a reduced form of a global solution.

$$b_{0,t} = \alpha_0, H^{-1}, \; K = M, H^{-1} \; and \; \mu_t = \varepsilon_t, H^{-1}$$

## 2.5 Impulse response analysis

The estimates of generalized impulse response functions (GIRFs) are based on the estimated GVAR model. The concept of GIRFs was suggested by Koop, Pesaran, and Potter [24] and further applied to the analysis of vector autoregressive (VAR) by Pesaran and Shin [25]. GIRF is different from the orthogonalized impulse response function (OIRF) proposed by Sims [26] for the dynamic analysis of the VAR model, which assumes shocks using Cholesky decomposition of the covariance matrix. This reduces the likelihood of errors. If a VAR consists of two or three variables, we can use orthogonalized impulse response functions (OIRFs). But the GVAR model has a large number of variables, so the OIRFs are not suitable. The advantage of GIRF is that it does not affect variables and countries. Thus, GIRF is determined by how shocks are transferred to variable "$j$" in the model, but the economic interpretation of these shocks is meaningless because variable "$j$" affects other different shocks. GIRF is useful for checking the GVAR model and the transmission of different shocks. Now in order for the interpretation of shocks here the generalized impulse response to a one standard error shock to the '$j$' variable in a country '$i$' at time '$t$' on the expected values of x at time $t+n$ are given.

Mathematically:

$$GIRF(x_t : V_{ilt}, n) = E[x_{t+h} | V_{ilt} = \sqrt{\infty_{ii,ll}}, \mathbb{I}_{t-1}] - E[x_{t+h} | \mathbb{I}_{t-1}] \tag{9}$$

Where $\infty_{ii,ll}$ is the corresponding diagonal element of the residual, variance-covariance matrix $\Sigma u$; $\mathbb{I}_{t-1}$ is the information set at time $t-1$.

## 2.6 Data and variables

This section describes the data and variables used in this study. Variables' definitions, their measurements, and data sources have been reported in Table 5.

## 3. Results and discussion

This section contains the estimated results of GVAR, which are based on different testing procedures. These include trade weight matrix, unit root test, selection of lag length criteria, co-

**Table 5. Variables' definitions and data source.**

| Variable | Symbol | Measurement | Data source |
|---|---|---|---|
| Real GDP | $Y_{it}$ | real $GDP_{it} = GDP_{it}/CPI_{it}$ | GVAR toolbox |
| Real equity price | $Q_{it}$ | real $equity\ price_{it} = Q_{it}/CPI_{it}$ | GVAR toolbox |
| Real exchange rate | $EX_{it}$ | $EX_{it}$ = Nominal $EX_{it}*CPI_{it}/CPI_{jt}$ | GVAR toolbox |
| Trade volume | $T_{it}$ | Trade volume of country $i$ | IMF |

Note: In our estimation, we have used quarterly data from 1990Q2 to 2016Q4 for four countries, which include China, the US, Japan, and Germany. The list of macroeconomic variables is provided in Table 1. The data supporting the study's findings are openly available in the GVAR toolbox at https://sites.google.com/site/gvarmodelling/data, with the reference number [GVAR Data 1979Q1-2016Q4].

**Table 6. Trade weight matrix.**

| Countries | China | Germany | Japan | US |
|---|---|---|---|---|
| China | 0 | 0.775 | 0.552 | 0.594 |
| Germany | 0.161 | 0 | 0.073 | 0.190 |
| Japan | 0.273 | 0.043 | 0 | 0.218 |
| US | 0.568 | 0.183 | 0.376 | 0 |

**Source**: Direction of Trade Statistics, 2014-2016, IMF.

**Note**: Trade weights are computed as shares of exports and imports displayed in column by region such that a column, but not a row, sums to one.

integration tests, weak exogeneity test, structural stability test, contemporaneous effect of a foreign variable on its domestic counterpart, and generalized impulse response function.

## 3.1 Trade weight matrix

While estimating the GVAR model, a trade weight matrix is required and constructed from the trade flow data given in Table 6. In such a case weights $w_{ij}$ is the country $i$ share in the trade of country $j$. Trade share matrix of the each country is constructed such that $w_{ii} = 0$ and $\Sigma w_{ij} = 1$.

Table 6 shows that four major trading partners have a 4×4 trade matrix. The trade of a country is the sum of its exports and imports. We have constructed a fixed trade weight matrix based on the average of three years' trade (such as 2014, 2015, and 2016) between two countries. In the individual country, shares are constructed by dividing the sum of total trade in the given period of each country $i$, by the amount of trade share with country $j$, such that the $i_{th}$ row sums to one for all $i$. Considering China, we can see that the Chinese trade share with Germany was 16.1%, Japan had 27.3%, and the remaining 56.8% was traded to the US.

## 3.2 Unit root test

The unit root test is applied to test the null hypothesis of non-stationarity at both the level and first difference of variables [27]. Table 7 shows that the domestic real GDP of Germany, Japan, and the US is non-stationary I (1), while in the case of China, the real GDP is also non-stationary even at the first difference. For Japan, the variable of equity prices is stationary at level i. e. I(0), and for the other countries, the series is stationary at first difference i. e. I(1). The domestic trade volume of China is found stationary at a level, i. e., I (0), while the others are I (1). In the case of the real exchange rate, China, Japan, and US data series are non-stationary, and the first-difference non-stationary in the case of Germany. The foreign real GDP for all countries is I (1) and so are real equity prices, also I (1). The foreign trade volume of China and Japan is I (1), while the trade volume of Germany and the United States is non-stationary at the 5% level of significance at the first difference. The exchange rate for all countries remains stationary at first difference that is I (1).

## 3.3 Selection of lag length for the VARX* model

In the VAR$X^*$ model, the selection of the lag order of domestic and foreign variables is not balanced. This study used Schwartz Bayesian criteria (SBC) for the selection of the lag order of domestic variables $p_i$ and also the selection of the lag order of foreign variables ($q_i$) The results are displayed in (Table 8), in which the optimal VAR$X^*$ orders for China, Germany, Japan, and the US are (2, 1), (2, 1), (2, 1), and (2, 1), respectively.

**Table 7. Unit root test for domestic variables.**

| Countries | China | Germany | Japan | US |
|---|---|---|---|---|
| $Y_{it}$ | -1.165 | -0.457 | -1.015 | -1.983 |
| $\Delta Y_{it}$ | $-2.296^a$ | -4.882 | -6.495 | -4.367 |
| $Q_{it}$ | -1.372 | -2.132 | $-3.402^{**}$ | -2.041 |
| $\Delta Q_{it}$ | -5.027 | -5.056 | N/A | -5.980 |
| $T_{it}$ | -1.571 | -1.995 | -1.212 | -1.835 |
| $\Delta T_{it}$ | $-1.513^a$ | -2.842 | -8.759 | -4.514 |
| $REX_{it}$ | -0.333 | -1.757 | -2.589 | -2.263 |
| $\Delta REX_{it}$ | -7.106 | $-1.956^a$ | -4.777 | -3.913 |
| $Y^*_{it}$ | -1.026 | -1.305 | -1.569 | -0.868 |
| $\Delta Y^*_{it}$ | -5.434 | -3.339 | -3.605 | -5.851 |
| $Q^*_{it}$ | -1.816 | -1.432 | -1.467 | -1.349 |
| $\Delta Q^*_{it}$ | -6.598 | -4.839 | -4.329 | -4.329 |
| $T^*_{it}$ | -1.590 | -1.647 | -1.724 | -1.681 |
| $\Delta T^*_{it}$ | -4.135 | $-1.672^a$ | -4.686 | $-1.899^a$ |
| $REX^*_{it}$ | -1.436 | -0.356 | -0.402 | -0.764 |
| $\Delta REX^*_{it}$ | -5.111 | -7.088 | -6.750 | -7.834 |

Note

** indicates significant at the 5% level of significance at the level of series, while 'a' indicates the acceptance of the null hypothesis on first difference series. The ADF statistics are based on univariate AR (p) specifications in the level of the variables with p≤ 5, and the statistics for the level and first differences of the variables are all computed based on the same sample period, namely, 1990Q2-2016Q4.

## 3.4 Co-integration test

Individual country co-integrating space rank is computed using trace statistics developed by Dickey and Fuller [28], which require integrated of order one and weakly exogenous independent variables. In Table 9, the trace test indicates one co-integrating relationship among the variables for all countries except China, which has two co-integrating vectors.

## 3.5 Testing weak exogeneity

As previously stated, the key estimating approach for $X^*_{it}$ in terms of the long run parameter is weak exogeneity. Table 10 indicates that, with the exception of Japan's real GDP, trade volume, and real exchange rate, the assumption of weak exogeneity is met for all nations' models. It means that the real GDP of Japan is influenced by the GDP of its trade partners but that the GDP of its trading partners is unaffected by Japanese GDP. Furthermore, while Japanese trade

**Table 8. Order of VARX\* model.**

| Countries | P | Q | Schwartz Bayesian Criteria (SBC) |
|---|---|---|---|
| China | 1 | 1 | 593.10 |
| China | 2 | 1 | 575.72 |
| Germany | 1 | 1 | 790.19 |
| Germany | 2 | 1 | 789.14 |
| Japan | 1 | 1 | 776.10 |
| Japan | 2 | 1 | 760.56 |
| US | 1 | 1 | 1177.9 |
| US | 2 | 1 | 1164.2 |

**Table 9. Trace statistic test for Co-integration.**

| Countries | China | Germany | Japan | US | CV 95% |
|---|---|---|---|---|---|
| r=0 | 149.41 | 111.08 | 137.61 | 95.74 | 87.61 |
| r=1 | 72.56 | 60.10** | 61.27** | 42.99** | 61.41 |
| r=2 | 34.27** | 24.86 | 25.27 | 20.29 | 38.95 |
| r=3 | 11.11 | 8.05 | 6.31 | 5.295 | 19.97 |

Note

** indicates significant co-integrating vector

volume is influenced by other trade volumes, its trading partners' trade volumes are unaffected. Other real exchange rates have little effect on the Japanese real exchange rate.

## 3.6 Structural stability test

In the context of a co-integrated model, structural stability is applicable to both short- and long-term coefficients and also to the error variance. As our interest is in discovering the transmission mechanisms between China and its major trading partners, we focus on the stability of short-run coefficients. Here we consider the country-specific error correction models of the residuals of the individual equations, which are based on structural stability tests. These residuals do not depend on the identification of co-integrating relations but depend on the rank of the co-integrating vectors. In our empirical analysis, different tests are included, such as the maximal OLS cumulative sum (CUSUM) statistic suggested by Ploberger and Krämer [29] and denoted by $PK_{sup}$ and its mean square variance by $PK_{msq}$. The $PK_{sup}$ statistic is similar to the CUSUM test originated by Brown, Durbin, and Evans [30], although the latter is based on recursive rather than OLS residuals to test the null hypothesis of parameter stability. As per the $PK_{sup}$ test which is reported in Table 11, the US real GDP is not stable at the 5% level of significance.

However, according to the $PK_{msq}$ test (reported in Table 12), all parameters are stable at a 5% level of significance except the US real GDP. The statistics based on parameter instability are computed through critical values of $PK_{msq}$, test, which are calculated from the GVAR (p) model solution using a boot-strap sample. Both versions of the tests of stability in parameters are rejected in at most 7 out of 34 cases. The tests have little statistical evidence to reject the null hypothesis that the coefficients are stable in the case of 90% boot-strap in our GVAR model.

## 3.7 Contemporaneous effects of foreign variables on their domestic counterpart

Table 13 shows that the contemporaneous effects of a foreign variable on domestic variables are collectively provided by the computed t-ratio using the Newey-West variance estimator.

**Table 10. Weak exogeneity of the country-specific foreign variables.**

| Countries | F-test | $Y_{it}$ | $Q_{it}$ | $T_{it}$ | $REX_{it}$ |
|---|---|---|---|---|---|
| China | 3.09 | 0.49 | 1.12 | 0.19 | 0.18 |
| Germany | 3.94 | 1.65 | 0.21 | 1.81 | 3.02 |
| Japan | 3.94 | 6.03** | 0.09 | 8.57** | 17.76** |
| US | 3.94 | 0.28 | 0.67 | 0.11 | - |

Note

** indicates significance at 5% level of significance.

**Table 11.** $PK_{sup}$ test statistics.

| Countries | $Y_{it}$ | $Q_{it}$ | $T_{it}$ | $REX_{it}$ |
|---|---|---|---|---|
| China | 0.81 (1.02) | 0.67 (1.16) | 0.49 (0.89) | 0.62 (0.94) |
| Germany | 0.84 (1.14) | 0.64 (1.16) | 0.91 (1.00) | 0.59 (1.34) |
| Japan | 0.79 (1.10) | 0.76 (1.16) | 0.55 (0.84) | 0.98 (1.33) |
| US | 1.58** (1.17) | 0.84 (1.10) | 0.42 (1.26) | 0.31 (1.06) |

Note

** indicates the rejection of the null hypothesis in which parameters are not stable at a 5% level of significance. The critical values in Figure (.) are the GVAR model-calculated critical values.

These estimates interpret the foreign variables' impact on domestic variables. Most of the expected elasticities have a positive sign, which is significant and informative regarding international transmission from foreign to domestic variables. For each country, positive values indicate that the domestic variables are over-affected by foreign variables and negative values indicate that domestic variables are unaffected by foreign variables.

To show the variables' relationship, we have seven assumptions, which are as follows:

Firstly, the foreign GDP has a significant impact on domestic GDP in the whole sample period. The hypothesis will be; $H_0 = \gamma_{i0,G,G^*} = 0$ VS $H_a = \gamma_{i0,G,G^*} \neq 0$.

Rejection of the null hypothesis leads to interdependence relationship between foreign GDP and domestic GDP.

Secondly, the foreign equity prices have a significant impact on domestic equity prices in the whole sample period; $H_0 = \gamma_{i0,eq,eq^*} = 0$ VS $H_a = \gamma_{i0,eq,eq^*} \neq 0$.

Rejection of the null hypothesis leads to interdependence relationship between foreign equity prices and domestic equity prices.

Thirdly, the foreign trade volume has a significant impact on domestic trade volume in the whole sample period; $H_0 = \gamma_{i0,tv,tv^*} = 0$ VS $H_a = \gamma_{i0,tv,tv^*} \neq 0$.

Rejection of the null hypothesis leads to interdependence relationship between foreign trade volume and domestic trade volume.

Fourthly, the foreign exchange rate has a significant impact on the domestic exchange rate in the whole sample period, $H_0 = \gamma_{i0,ex,ex^*} = 0$ VS $H_a = \gamma_{i0,ex,ex^*} \neq 0$

Rejection of the null hypothesis leads to interdependence relationship between foreign exchange rate and domestic exchange rate.

These elasticities are very informative about the international linkages between the domestic and foreign variables. Table 13 shows that, focusing on the Chinese economy, a 1% change in foreign real GDP in a given period leads to an increase of 0.5% in Chinese real GDP, but the effect is insignificant within the same period. Similarly, foreign real GDP elasticities are

**Table 12.** $PK_{msq}$ test statistics.

| Country | $Y_{it}$ | $Q_{it}$ | $T_{it}$ | $REX_{it}$ |
|---|---|---|---|---|
| China | 0.14 (0.28) | 0.10 (0.34) | 0.05 (0.18) | 0.03 (0.20) |
| Germany | 0.14 (0.36) | 0.07 (0.38) | 0.09 (0.25) | 0.04 (0.46) |
| Japan | 0.08 (0.34) | 0.18 (0.28) | 0.05 (0.11) | 0.12 (0.53) |
| US | 0.68** (0.41) | 0.11 (0.26) | 0.03 (0.46) | 0.01 (0.26) |

Note

** indicates the rejection of the null hypothesis in which parameters are not stable at a 5% level of significance. The critical values in Figure (.) are the GVAR model-calculated critical values.

**Table 13. Contemporaneous effect of foreign variables on domestic variables.**

| Countries | GDP | Equity Prices | Trade Volume | Exchange Rate |
|---|---|---|---|---|
| China | 0.50 (1.88) | 0.37 (1.29) | 2.76 (8.69)* | -0.01 (-0.08) |
| Germany | 0.04 (0.55) | 0.16 (1.68) | 0.12 (4.33)* | 0.17 (0.92) |
| Japan | 0.20 (1.73) | 0.16 (0.97) | 0.40 (7.13)* | -0.11 (-0.60) |
| US | 0.02 (0.36) | 0.16 (1.94)** | 0.29 (7.23)* | N/A |

Note

* indicates significant at 1% level of significance and ** indicates significant at 5% level of significance

obtained from the different regions because the effect is slightly weaker for Germany, Japan, and the US. In the case of the Chinese economy, the elasticity estimate is relatively large but insignificant. More importantly, the contemporaneous elasticity of real equity prices in China is greater than the elasticities of other economies. Hence, the Chinese stock market seems to overreact to foreign stock prices, but the effect is insignificant. Similar results were obtained for Germany and Japan, but they remained insignificant. The effects of US financial linkage are strong and significant. We can also observe the strong co-movement between China and foreign trade volume. And the remaining economies, like Germany, Japan, and the US, are also positively affected by foreign trade volume, which is significant for all countries. In contrast, Germany's exchange rate is positively affected by the foreign exchange rate and the remaining economy, Japan's exchange rate is not much affected by the foreign exchange rate, and China's exchange rate is slightly negatively affected by the foreign exchange rate.

## 3.8 Generalized impulse response functions

To study the dynamic properties of the global model and to measure the time profile of the Chinese economy's effects on macroeconomic variables of its major trading partners' economies, we investigate the effects of a one-standard-error positive shock on Chinese real GDP, stock prices, trade volume, and the exchange rate. Here we use the generalized impulse response to the estimated GVAR model, which was proposed by Ploberger and Krämer [29] and further added a new option by Koop et al. [24]. The GIRF is the more appropriate dynamic for the transfer of shocks from China to its major trading partners. In Fig 2, a one-standard-error positive shock to Chinese real GDP is given to trace the profile of its impact on the macroeconomic variables of the trading partners. The red line shows the significant impact. It can be seen from Fig 2.

China's GDP shock boosts Germany's GDP and total trade while also appreciating its currency over time. However, it has a negligible impact on its stock prices. The same results appeared for Japan. In the case of the US, it worsens the US's GDP and its stock prices but appreciates its currency. Therefore, it is concluded that the shock to Chinese GDP has an insignificant impact on the macroeconomic variables of Germany. Japan will enjoy the shock of Chinese GDP by improving its own. However, it appreciates the US currency only among its macroeconomic variables and will worsen its GDP.

The findings of our study are consistent with Chen, Huang, and Huang [31], who found that a negative shock in Japan's real output causes an immediate fall in Taiwan's and Korea's real GDP. However, these impacts quickly become positive from quarters 3 through 9. Furthermore, following quarter 9, Taiwan's real GDP falls to a negative value, while Korean real GDP remains positive. Japan is not just Taiwan's largest trading partner but also a major supplier of materials and components to important sectors in Taiwan. If Japan's real output suffers an external negative shock, Taiwan's real GDP also suffers as a result of this strong economic

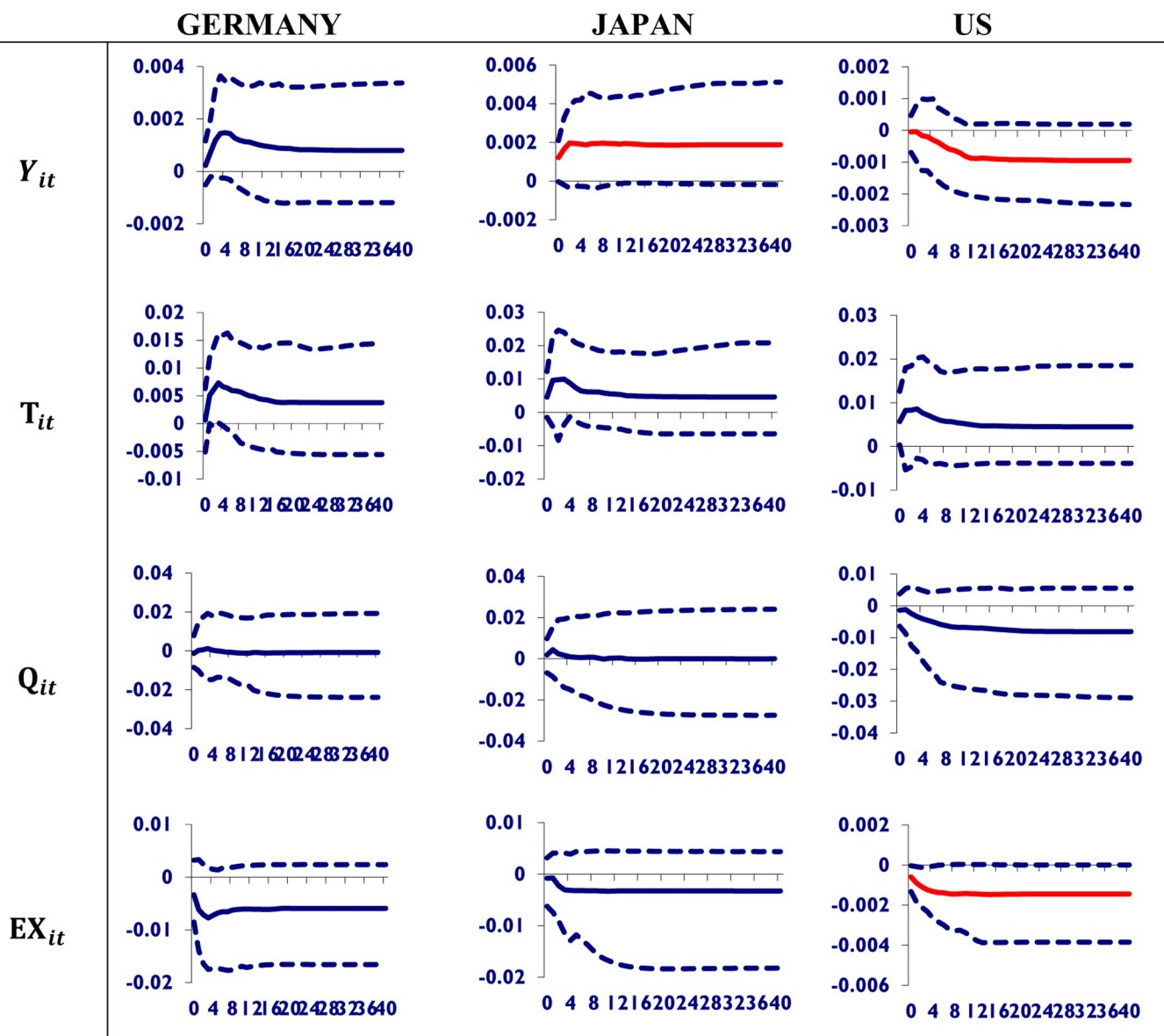

**Fig 2. Shocks to Chinese real GDP and its impact on trading partners.** A shock to Chinese real GDP has an insignificant impact on the macroeconomic variables of Germany. Japan will enjoy the shock of Chinese GDP by improving its own. However, it appreciates the US currency only among its macroeconomic variables and will worsen its GDP.

connection. A decline in US real GDP may imply a decrease in the real output of many countries around the world, as well as fewer job opportunities in countries whose economies are heavily dependent on trade with the US, such as Taiwan. As shown in the middle-right panel of Fig 2, a decline in US real GDP reduces output in almost all selected foreign countries, with the effects being especially pronounced in US neighboring countries such as Mexico and Canada. Furthermore, the impacts are severe for some countries, particularly the smaller Asian economies (e.g., Taiwan, Hong Kong, and Singapore). Large Asian countries like Japan, Korea, and China appear to be relatively unscathed.

China's output shock, on the other hand, has no effect on macroeconomic activity in Taiwan, Korea, or the rest of the world. Taiwan and the majority of selected countries have limited

and insignificant effects on real output, real equity prices, and interest rates. These findings could be explained by the fact that China has recently experienced rapid growth. Our study only used data up to 2008: QIV, which may not be long enough to capture the impact of a shock to macroeconomic factors in China on the world as significantly as one in the United States. According to Dinda's [32] research, the People's Republic of China's (PRC) slowdown has a greater impact on emerging BRICS economies than on developed economies. The PRC's GDP growth shock has had a limited impact on other developing economies. In terms of the global linkage variable, the Chinese economic slowdown has a direct impact on emerging and developed economies.

In Fig 3, a one-standard-error positive shock to Chinese trade volume is given. It can be seen that the shock to Chinese trade a little bit improves the GDP and total trade volume of Germany and has an insignificant impact on its currency and its stock prices. It deteriorates Japan's GDP and total trade volume. Whereas the US is more vulnerable to shocks from Chinese trade because it decreases its trade volume and depreciates its currency permanently, Germany's economy is resilient to the trade shock from China. According to Zhang and Gui [33], China's trade production has strong network effects as a result of Asia's growing integrated regional production system. Such an integrated system has boosted exports from these countries to China as well as overall exports to third markets. With the exception of Mexico, China's economy has no significant impact on trade in Latin America and Africa. Countries with export structures similar to China's have experienced competitive pressure from China, whereas countries with import structures similar to China's have experienced complementary effects from China's economic development.

Fig 4 shows that a shock to Chinese equity prices reduces GDP, trade volume, and strengthens Germany's currency while having little effect on stock prices. In the case of Japan, this shock deteriorates its GDP, which is insignificant to its trade volume, increases equity prices, and depreciates the currency. It significantly appreciates the US currency and lowers its GDP. Therefore, it is concluded that the shock to Chinese equity prices is more alarming for the German economy than the Chinese one because it permanently reduces its GDP, deteriorates its trade volume, appreciates its currency, and improves its stock prices. The Japanese economy seems to be doing well in terms of its GDP improvement. It also significantly appreciates the US currency. According to Shi [34], for the group of developed partners, the Chinese stock market is more correlated with Asian markets than with Pacific markets, and for the group of emerging partners, it is more integrated with East Asian markets. Although different pair-wise correlations reveal different evolving patterns, most pair-wise co-movements show increasing trends between mid-2015 and early-2018. Second, contagion episodes suggest that turbulent events amplify stock market co-movements, with levels of contagion frequency increasing during the Shanghai stock market crash, US-China tariff wars, and COVID-19 pandemic. However, the findings for developed and emerging economies differ slightly.

There are signs that stock market co-movement between China and its developed partners is more sensitive to US-China trade frictions than stock market co-movement between China and its emerging partners, which is more influenced by the Shanghai stock market crash. Third, the regression results reveal several regularities in the drivers of stock market co-movement. According to Shu, He, Wang, and Dong [35], who examined China's impact on Asian-Pacific financial markets, the analysis shows that China has a significant influence on regional equity and currency markets. In normal or non-stress times, China's influence on Asian stock markets approaches that of the US, though spillovers from the US to Asian financial markets tend to be stronger than China's influence in stressful times. Following China's transition to a managed float regime in 2005, the RMB has also gained the ability to move regional currencies. Nonetheless, China's bond market remains isolated from those of the rest of Asia and the US.

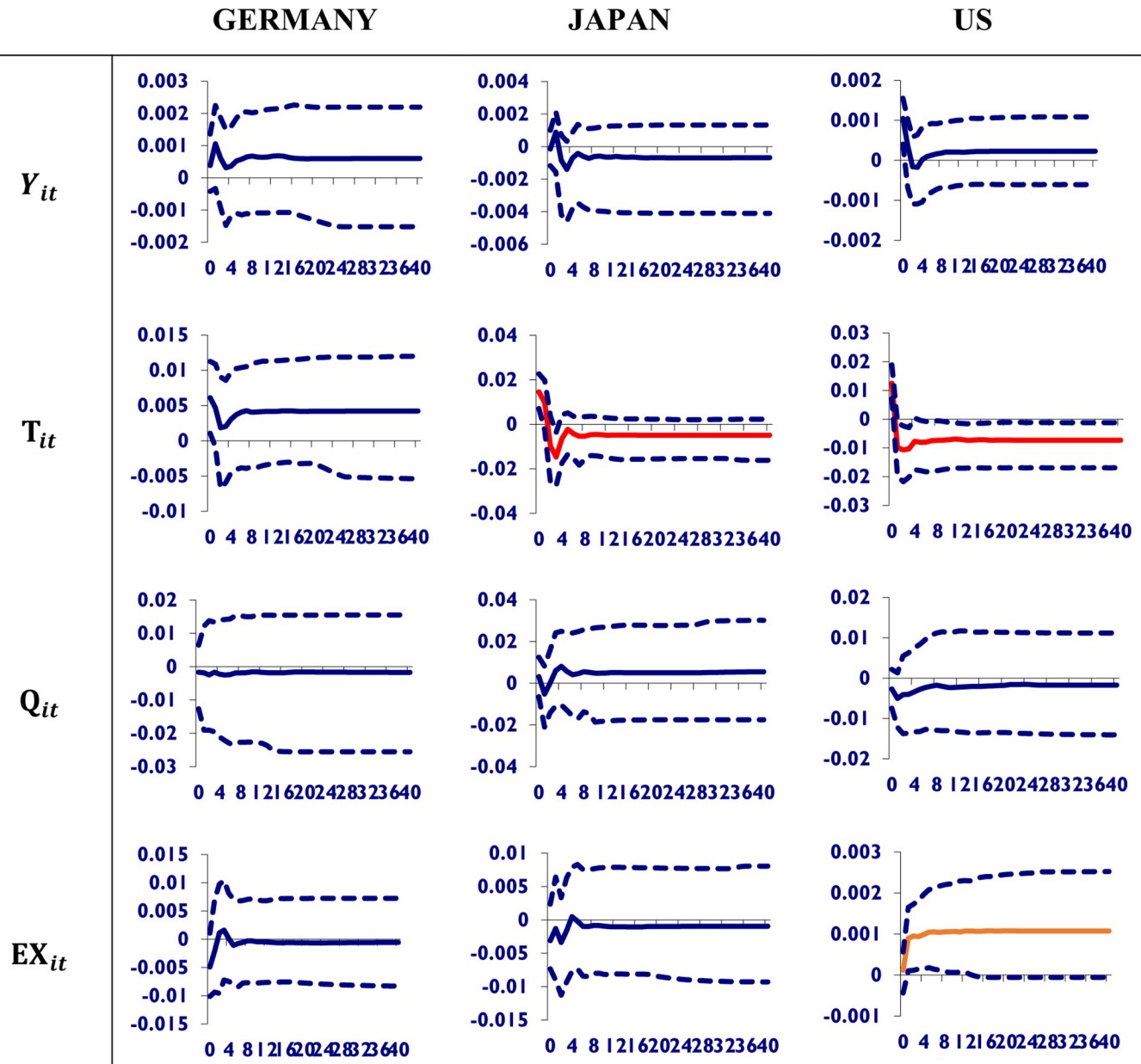

**Fig 3. Shocks to Chinese trade volume and its impact on its trading partners economies.** A shock to Chinese trade volume has a little bit improved the GDP and total trade volume of Germany and has had an insignificant impact on its currency and its stock prices. It deteriorates Japan's GDP and total trade volume. Whereas the US is more vulnerable to shocks from Chinese trade because it decreases its trade volume and depreciates its currency permanently, Germany's economy is resilient to the trade shock from China.

China's growing influence, combined with that of the United States, has significant implications for Asian financial markets and capital flows. Then the United States' influence on global markets remains dominant, driving global liquidity conditions and risk appetite, especially during times of stress.

Fig 5 indicates the impact of a shock to the Chinese exchange rate on the macroeconomic variables of its trading partners. It is clearly evident that the shock to the Chinese exchange rate is good for Germany as it improves its GDP and trade volume, but bad for the US as it

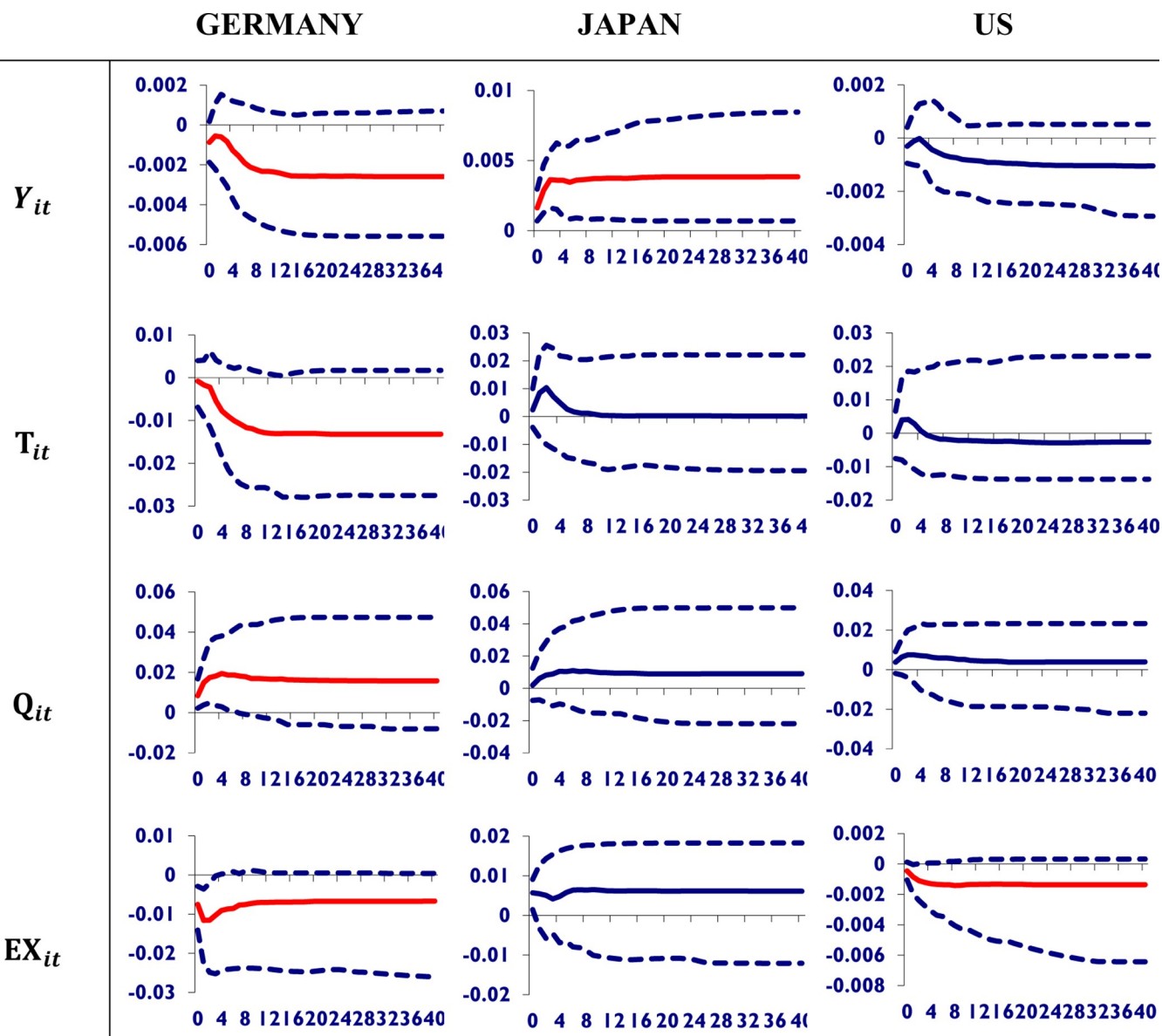

**Fig 4. Shocks to Chinese stock prices and its impact on its trading partners economies.** A shock to Chinese equity prices is more alarming for the German economy than the Chinese one because it permanently reduces its GDP, deteriorates its trade volume, appreciates its currency, and improves its stock prices. The Japanese economy seems to be doing well in terms of its GDP improvement. It also significantly appreciates the US currency.

worsens its equity prices. However, Japan's currency will be stronger in terms of appreciation due to the shock in the Chinese exchange rate.

According to Ahmed et al. [14], an appreciation of the Chinese currency should unambiguously reduce both Chinese processing and non-processing exports to its trade partners with advanced economies. Our empirical results are generally consistent with these predictions. We found a significant negative effect of RMB appreciation on China's exports to advanced economies. However, the impact of appreciation was found to be positive on the exports of processed goods to emerging Asian economies. But this finding was statistically insignificant. Furthermore, the impact of global imbalances is dependent on how they are defined, which is not always clear. There is substantial disagreement over the proxies used to quantify global

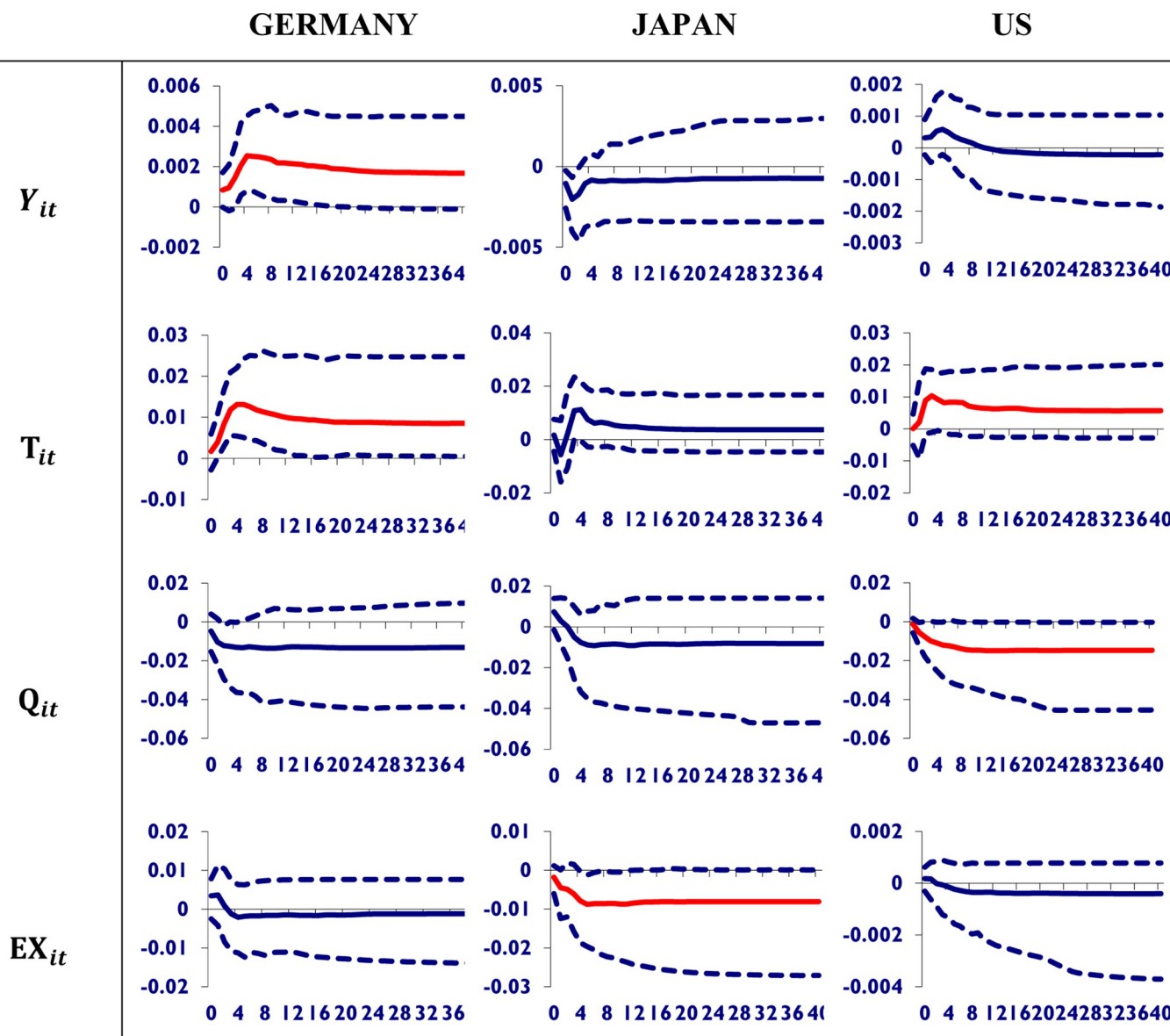

**Fig 5. Shocks to Chinese real exchange rate and its impact on its trading partners economies.** A shock to the Chinese exchange rate is good for Germany as it improves its GDP and trade volume, but bad for the US as it worsens its equity prices. However, Japan's currency will be stronger in terms of appreciation due to the shock in the Chinese exchange rate.

imbalances, and the findings may vary as these proxies change. Our findings suggest that if China's large current account surplus or bilateral current account surplus with the US, along with the US bilateral current account deficit with China, contributes to global imbalances, a greater degree of Chinese currency appreciation would significantly help mitigate global imbalances. If, on the other hand, the US overall current account deficit and the emerging market world's current account surplus are the major components of global imbalances, then it is less evident that further Chinese currency appreciation would put a significant dent in global imbalances.

The existing literature shows that the exchange rate of China has a direct influence on its trading partners. China's trading partners' exports have suffered as a result of the currency's

devaluation. Since China has a managed float exchange rate system, it allows a sovereign central bank to regularly intervene in the market to modify the direction of the currency float and boost its balance of payments in completely impulsive phases. As a result, Chinese exports are rapidly rising. Besides this, other initiatives in the form of BRI may further expand the volume of trade with major trade partners. Aside from this, additional BRI initiatives may boost the volume of trade with major trading partners. The GVAR approach is preferred for capturing impacts in such global dynamic settings [31]. As a result, we investigated the influence of Chinese macroeconomic performance, particularly the exchange rate, on global trade. The survey of literature also indicated that China's exchange rate is complicated, necessitating a more in-depth inquiry of trade linkages with currency rates as well as an evaluation of risk and uncertainty for its trading partners. Table 4 presents the findings of various studies on China's currency devaluation and trade policies, as well as their influence on trading partners, with conflicting results showing a lack of consensus. Understanding China's interactions with its trading partners is crucial for developing successful trade policies that reduce the effects of Chinese economic shocks. As a result, the current study is intended to respond to shocks in China's macroeconomic variables and their impact on its trading partners.

## 4. Conclusion and policy implications

The aim of the article is to quantify the impact of changes in macroeconomic factors on China's trading partners (Germany, Japan, and the US). It is apparent that the Chinese economy has an asymmetric impact on its trading partners. For example, Germany should not be concerned about China's rising GDP and trade volume, but rather adopt measures in its economy to maintain its GDP and trade volume as China's share prices rise. Germany benefits from the depreciation of the Yuan since it boosts its GDP and trade volume. The Japanese economy is the least influenced by the Chinese economy since a shock to Chinese GDP and trade volume boosts Japan's GDP, and a devaluation of the Chinese currency improves its currency. However, as China's trade volume grows, Japan should be concerned since this would diminish the country's GDP. According to the findings, the United States is the trading partner most exposed to increases in Chinese GDP, trade volume, and exchange rates. An increase in Chinese trade volume would result in a major reduction in trade volume and a depreciation of the Chinese currency. Furthermore, the Yuan's devaluation will have a huge impact on US stock prices. Based on these findings, it may be inferred that the US economic forecast is accurate. As a result, taxes on Chinese products are defensive in nature in order to defend the US economy. Another contentious subject that may be addressed using the GVAR model is the impact of the US-China trade war on the global economy.

Over the last two decades, China has risen to become one of the world's top three exporters, with key trading partners including the United States, Germany, and Japan. Furthermore, China's BRI and the formation of regional trade agreements (RTAs) are expected to boost China's economic performance and trade around the world, having a significant impact on the global economy. The former US President and Chairman of the Federal Reserve of the United States denoted the Chinese currency as artificially weak and blamed Chinese authorities for artificially keeping the Yuan low in order to attract global demand and maintain a trade advantage. As a result, such an issue has sparked heated debate in diplomatic and political circles. Furthermore, numerous studies have been conducted, resulting in significant disagreement among researchers. The optimality of the Yuan exchange rate and its valuation have been frequently studied, with the main focus being on determining whether the Chinese currency is undervalued and, if so, by how much [36–38].

Furthermore, one line of research looked into the Yuan's optimal adjustments [39] and possible Chinese government responses to alleviate currency appreciation pressure [40]. Though

currency appreciation pressure is primarily driven by China's trade imbalance with its major trading partners, little research has been conducted on the impact of the Yuan's exchange rate on trade. It is worth noting that China's managed float exchange rate policy denotes a situation in which Chinese exports to trading partners exceed imports from them. As a result, a fluctuation in the exchange rate may cause China's trade balance to fluctuate further. Another important variable that can be examined using the GVAR impulse response function is the trade balance of trading partners. This investigation can assist policymakers and authorities in China's trading partners in designing policy structures to reduce their trade deficits and realize trade gains.

## Supporting information

**S1 Data. Data selected variables used in GVAR.**
(XLSX)

## Author Contributions

**Conceptualization:** Aftab Alam.

**Formal analysis:** Ibrar Hussain.

**Investigation:** Jingmei Ma.

**Methodology:** Aftab Alam, Rizwan Fazal.

**Software:** Aftab Alam.

**Supervision:** Jingmei Ma.

**Validation:** Jingmei Ma.

**Visualization:** Jingmei Ma.

**Writing – original draft:** Aftab Alam.

**Writing – review & editing:** Rizwan Fazal.

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
