## [Decision Letter · Decision Letter 0]

20 Jul 2022

PONE-D-22-13263An analysis of the impact of China’s macroeconomic performance on its trade partners: Evidence based on the GVAR modelPLOS ONE

Dear Dr. Hussain,

Thank you for submitting your manuscript to PLOS ONE. After careful consideration, we feel that it has merit but does not fully meet PLOS ONE’s publication criteria as it currently stands. Therefore, we invite you to submit a revised version of the manuscript that addresses the points raised during the review process. Specifically, the paper should underscore the value added and the relationship with the existing literature.

We look forward to receiving your revised manuscript.

Kind regards,

Petre Caraiani

Academic Editor

PLOS ONE

Journal Requirements:

Reviewers' comments:

Reviewer's Responses to Questions

**Comments to the Author**

1. Is the manuscript technically sound, and do the data support the conclusions?

Reviewer #1: Yes

2. Has the statistical analysis been performed appropriately and rigorously? 

Reviewer #1: Yes

3. Have the authors made all data underlying the findings in their manuscript fully available?

Reviewer #1: Yes

4. Is the manuscript presented in an intelligible fashion and written in standard English?

Reviewer #1: Yes

5. Review Comments to the Author

Reviewer #1: The paper is fair enough from the point of view of metholodogy and data analysis, however I think the authors should emphasize the added value of the paper.

What is the novelty of the paper and how the results are in line with the existing literature?

6. PLOS authors have the option to publish the peer review history of their article (what does this mean?). If published, this will include your full peer review and any attached files.

Reviewer #1: No

---

## [Author Response · Author response to Decision Letter 0]

18 Aug 2022

1. The paper has been revised and emphasis been made on the value added in section 3.8 and section 4. Besides this, the novelty of the paper and the findings of our study have been compared and contrasted with the existing literature in section 3.8.

---

## [Decision Letter · Decision Letter 1]

26 Sep 2022

An analysis of the impact of China’s macroeconomic performance on its trade partners: Evidence based on the GVAR model

PONE-D-22-13263R1

Dear Dr. Ma,

We’re pleased to inform you that your manuscript has been judged scientifically suitable for publication and will be formally accepted for publication once it meets all outstanding technical requirements.

Kind regards,

Petre Caraiani

Academic Editor

PLOS ONE

Additional Editor Comments (optional):

The manuscript should be edited in terms of language.

Reviewers' comments:

Reviewer's Responses to Questions

**Comments to the Author**

1. If the authors have adequately addressed your comments raised in a previous round of review and you feel that this manuscript is now acceptable for publication, you may indicate that here to bypass the “Comments to the Author” section, enter your conflict of interest statement in the “Confidential to Editor” section, and submit your "Accept" recommendation.

Reviewer #1: All comments have been addressed

2. Is the manuscript technically sound, and do the data support the conclusions?

Reviewer #1: Yes

3. Has the statistical analysis been performed appropriately and rigorously? 

Reviewer #1: Yes

4. Have the authors made all data underlying the findings in their manuscript fully available?

Reviewer #1: Yes

5. Is the manuscript presented in an intelligible fashion and written in standard English?

Reviewer #1: Yes

6. Review Comments to the Author

Reviewer #1: The authors adressed all the comments and suggestions. I think that in the present form the manuscript is suitable for publication.

7. PLOS authors have the option to publish the peer review history of their article (what does this mean?). If published, this will include your full peer review and any attached files.

Reviewer #1: No

---

## [Editor Report · Acceptance letter]

6 Oct 2022

PONE-D-22-13263R1 

An analysis of the impact of China’s macroeconomic performance on its trade partners: Evidence based on the GVAR model 

Dear Dr. Ma:

I'm pleased to inform you that your manuscript has been deemed suitable for publication in PLOS ONE. Congratulations! Your manuscript is now with our production department. 

Kind regards, 

on behalf of

Dr. Petre Caraiani 

Academic Editor

PLOS ONE